# Deep Generative Image Models using a Laplacian Pyramid of Adversarial Networks

**Emily Denton**$^*$  **Soumith Chintala**$^*$  **Arthur Szlam**  **Rob Fergus**

Dept. of Computer Science
Courant Institute
New York University

Facebook AI Research
New York

## Abstract

In this paper we introduce a generative parametric model capable of producing high quality samples of natural images. Our approach uses a cascade of convolutional networks within a Laplacian pyramid framework to generate images in a coarse-to-fine fashion. At each level of the pyramid, a separate generative convnet model is trained using the Generative Adversarial Nets (GAN) approach [11]. Samples drawn from our model are of significantly higher quality than alternate approaches. In a quantitative assessment by human evaluators, our CIFAR10 samples were mistaken for real images around 40% of the time, compared to 10% for samples drawn from a GAN baseline model. We also show samples from models trained on the higher resolution images of the LSUN scene dataset.

## 1   Introduction

Building a good generative model of natural images has been a fundamental problem within computer vision. However, images are complex and high dimensional, making them hard to model well, despite extensive efforts. Given the difficulties of modeling entire scene at high-resolution, most existing approaches instead generate image patches. In contrast, we propose an approach that is able to generate plausible looking scenes at $32 \times 32$ and $64 \times 64$. To do this, we exploit the multi-scale structure of natural images, building a series of generative models, each of which captures image structure at a particular scale of a Laplacian pyramid [1]. This strategy breaks the original problem into a sequence of more manageable stages. At each scale we train a convolutional network-based generative model using the Generative Adversarial Networks (GAN) approach of Goodfellow *et al.* [11]. Samples are drawn in a coarse-to-fine fashion, commencing with a low-frequency residual image. The second stage samples the band-pass structure at the next level, conditioned on the sampled residual. Subsequent levels continue this process, always conditioning on the output from the previous scale, until the final level is reached. Thus drawing samples is an efficient and straightforward procedure: taking random vectors as input and running forward through a cascade of deep convolutional networks (convnets) to produce an image.

Deep learning approaches have proven highly effective at discriminative tasks in vision, such as object classification [4]. However, the same level of success has not been obtained for generative tasks, despite numerous efforts [14, 26, 30]. Against this background, our proposed approach makes a significant advance in that it is straightforward to train and sample from, with the resulting samples showing a surprising level of visual fidelity.

### 1.1   Related Work

Generative image models are well studied, falling into two main approaches: non-parametric and parametric. The former copy patches from training images to perform, for example, texture synthesis [7] or super-resolution [9]. More ambitiously, entire portions of an image can be in-painted, given a sufficiently large training dataset [13]. Early parametric models addressed the easier problem of tex-

---

$^*$denotes equal contribution.

ture synthesis [3, 33, 22], with Portilla & Simoncelli [22] making use of a steerable pyramid wavelet representation [27], similar to our use of a Laplacian pyramid. For image processing tasks, models based on marginal distributions of image gradients are effective [20, 25], but are only designed for image restoration rather than being true density models (so cannot sample an actual image). Very large Gaussian mixture models [34] and sparse coding models of image patches [31] can also be used but suffer the same problem.

A wide variety of deep learning approaches involve generative parametric models. Restricted Boltzmann machines [14, 18, 21, 23], Deep Boltzmann machines [26, 8], Denoising auto-encoders [30] all have a generative decoder that reconstructs the image from the latent representation. Variational auto-encoders [16, 24] provide probabilistic interpretation which facilitates sampling. However, for all these methods convincing samples have only been shown on simple datasets such as MNIST and NORB, possibly due to training complexities which limit their applicability to larger and more realistic images.

Several recent papers have proposed novel generative models. Dosovitskiy *et al.* [6] showed how a convnet can draw chairs with different shapes and viewpoints. While our model also makes use of convnets, it is able to sample general scenes and objects. The DRAW model of Gregor *et al.* [12] used an attentional mechanism with an RNN to generate images via a trajectory of patches, showing samples of MNIST and CIFAR10 images. Sohl-Dickstein *et al.* [28] use a diffusion-based process for deep unsupervised learning and the resulting model is able to produce reasonable CIFAR10 samples. Theis and Bethge [29] employ LSTMs to capture spatial dependencies and show convincing inpainting results of natural textures.

Our work builds on the GAN approach of Goodfellow *et al.* [11] which works well for smaller images (e.g. MNIST) but cannot directly handle large ones, unlike our method. Most relevant to our approach is the preliminary work of Mirza and Osindero [19] and Gauthier [10] who both propose conditional versions of the GAN model. The former shows MNIST samples, while the latter focuses solely on frontal face images. Our approach also uses several forms of conditional GAN model but is much more ambitious in its scope.

## 2  Approach

The basic building block of our approach is the generative adversarial network (GAN) of Goodfellow *et al.* [11]. After reviewing this, we introduce our LAPGAN model which integrates a conditional form of GAN model into the framework of a Laplacian pyramid.

### 2.1  Generative Adversarial Networks

The GAN approach [11] is a framework for training generative models, which we briefly explain in the context of image data. The method pits two networks against one another: a generative model $G$ that captures the data distribution and a discriminative model $D$ that distinguishes between samples drawn from $G$ and images drawn from the training data. In our approach, both $G$ and $D$ are convolutional networks. The former takes as input a noise vector $z$ drawn from a distribution $p_{\text{Noise}}(\mathbf{z})$ and outputs an image $\tilde{h}$. The discriminative network $D$ takes an image as input stochastically chosen (with equal probability) to be either $\tilde{h}$ – as generated from $G$, or $h$ – a real image drawn from the training data $p_{\text{Data}}(\mathbf{h})$. $D$ outputs a scalar probability, which is trained to be high if the input was real and low if generated from $G$. A minimax objective is used to train both models together:

$$\min_G \max_D \mathbb{E}_{h \sim p_{\text{Data}}(\mathbf{h})}[\log D(h)] + \mathbb{E}_{z \sim p_{\text{Noise}}(\mathbf{z})}[\log(1 - D(G(z)))] \tag{1}$$

This encourages $G$ to fit $p_{\text{Data}}(\mathbf{h})$ so as to fool $D$ with its generated samples $\tilde{h}$. Both $G$ and $D$ are trained by backpropagating the loss in Eqn. 1 through both models to update the parameters.

The conditional generative adversarial net (CGAN) is an extension of the GAN where both networks $G$ and $D$ receive an additional vector of information $l$ as input. This might contain, say, information about the class of the training example $h$. The loss function thus becomes

$$\min_G \max_D \mathbb{E}_{h,l \sim p_{\text{Data}}(\mathbf{h},\mathbf{l})}[\log D(h, l)] + \mathbb{E}_{z \sim p_{\text{Noise}}(\mathbf{z}), l \sim p_l(\mathbf{l})}[\log(1 - D(G(z, l), l))] \tag{2}$$

where $p_l(\mathbf{l})$ is, for example, the prior distribution over classes. This model allows the output of the generative model to be controlled by the conditioning variable $l$. Mirza and Osindero [19] and Gauthier [10] both explore this model with experiments on MNIST and faces, using $l$ as a class indicator. In our approach, $l$ will be another image, generated from another CGAN model.

## 2.2 Laplacian Pyramid

The Laplacian pyramid [1] is a linear invertible image representation consisting of a set of band-pass images, spaced an octave apart, plus a low-frequency residual. Formally, let $d(.)$ be a downsampling operation which blurs and decimates a $j \times j$ image $I$, so that $d(I)$ is a new image of size $j/2 \times j/2$. Also, let $u(.)$ be an upsampling operator which smooths and expands $I$ to be twice the size, so $u(I)$ is a new image of size $2j \times 2j$. We first build a Gaussian pyramid $\mathcal{G}(I) = [I_0, I_1, \ldots, I_K]$, where $I_0 = I$ and $I_k$ is $k$ repeated applications of $d(.)$ to $I$, i.e. $I_2 = d(d(I))$. $K$ is the number of levels in the pyramid, selected so that the final level has very small spatial extent ($\leq 8 \times 8$ pixels).

The coefficients $h_k$ at each level $k$ of the Laplacian pyramid $\mathcal{L}(I)$ are constructed by taking the difference between adjacent levels in the Gaussian pyramid, upsampling the smaller one with $u(.)$ so that the sizes are compatible:

$$h_k = \mathcal{L}_k(I) = \mathcal{G}_k(I) - u(\mathcal{G}_{k+1}(I)) = I_k - u(I_{k+1}) \tag{3}$$

Intuitively, each level captures image structure present at a particular scale. The final level of the Laplacian pyramid $h_K$ is not a difference image, but a low-frequency residual equal to the final Gaussian pyramid level, i.e. $h_K = I_K$. Reconstruction from a Laplacian pyramid coefficients $[h_1, \ldots, h_K]$ is performed using the backward recurrence:

$$I_k = u(I_{k+1}) + h_k \tag{4}$$

which is started with $I_K = h_K$ and the reconstructed image being $I = I_o$. In other words, starting at the coarsest level, we repeatedly upsample and add the difference image $h$ at the next finer level until we get back to the full resolution image.

## 2.3 Laplacian Generative Adversarial Networks (LAPGAN)

Our proposed approach combines the conditional GAN model with a Laplacian pyramid representation. The model is best explained by first considering the sampling procedure. Following training (explained below), we have a set of generative convnet models $\{G_0, \ldots, G_K\}$, each of which captures the distribution of coefficients $h_k$ for natural images at a different level of the Laplacian pyramid. Sampling an image is akin to the reconstruction procedure in Eqn. 4, except that the generative models are used to produce the $h_k$'s:

$$\tilde{I}_k = u(\tilde{I}_{k+1}) + \tilde{h}_k = u(\tilde{I}_{k+1}) + G_k(z_k, u(\tilde{I}_{k+1})) \tag{5}$$

The recurrence starts by setting $\tilde{I}_{K+1} = 0$ and using the model at the final level $G_K$ to generate a residual image $\tilde{I}_K$ using noise vector $z_K$: $\tilde{I}_K = G_K(z_K)$. Note that models at all levels except the final are conditional generative models that take an upsampled version of the current image $\tilde{I}_{k+1}$ as a conditioning variable, in addition to the noise vector $z_k$. Fig. 1 shows this procedure in action for a pyramid with $K = 3$ using 4 generative models to sample a $64 \times 64$ image.

The generative models $\{G_0, \ldots, G_K\}$ are trained using the CGAN approach at each level of the pyramid. Specifically, we construct a Laplacian pyramid from each training image $I$. At each level we make a stochastic choice (with equal probability) to either (i) construct the coefficients $h_k$ either using the standard procedure from Eqn. 3, or (ii) generate them using $G_k$:

$$\tilde{h}_k = G_k(z_k, u(I_{k+1})) \tag{6}$$

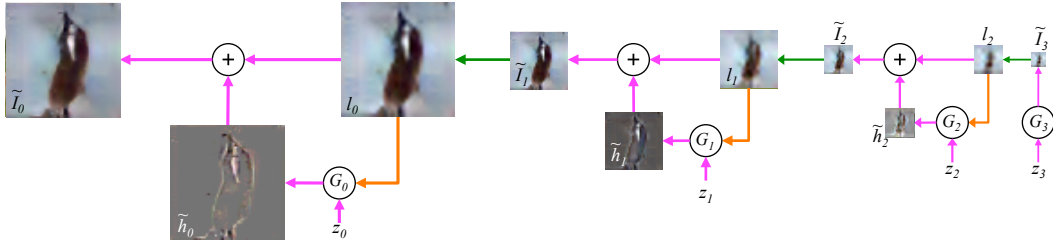

Figure 1: The sampling procedure for our LAPGAN model. We start with a noise sample $z_3$ (right side) and use a generative model $G_3$ to generate $\tilde{I}_3$. This is upsampled (green arrow) and then used as the conditioning variable (orange arrow) $l_2$ for the generative model at the next level, $G_2$. Together with another noise sample $z_2$, $G_2$ generates a difference image $\tilde{h}_2$ which is added to $l_2$ to create $\tilde{I}_2$. This process repeats across two subsequent levels to yield a final full resolution sample $I_0$.

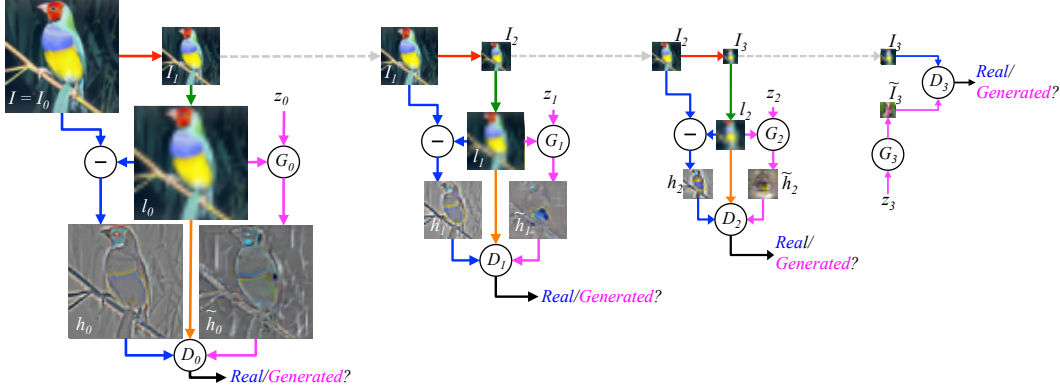

Figure 2: The training procedure for our LAPGAN model. Starting with a 64x64 input image $I$ from our training set (top left): (i) we take $I_0 = I$ and blur and downsample it by a factor of two (red arrow) to produce $I_1$; (ii) we upsample $I_1$ by a factor of two (green arrow), giving a low-pass version $l_0$ of $I_0$; (iii) with equal probability we use $l_0$ to create *either* a real *or* a generated example for the discriminative model $D_0$. In the real case (blue arrows), we compute high-pass $h_0 = I_0 - l_0$ which is input to $D_0$ that computes the probability of it being real vs generated. In the generated case (magenta arrows), the generative network $G_0$ receives as input a random noise vector $z_0$ and $l_0$. It outputs a generated high-pass image $\tilde{h}_0 = G_0(z_0, l_0)$, which is input to $D_0$. In both the real/generated cases, $D_0$ also receives $l_0$ (orange arrow). Optimizing Eqn. 2, $G_0$ thus learns to generate realistic high-frequency structure $\tilde{h}_0$ consistent with the low-pass image $l_0$. The same procedure is repeated at scales 1 and 2, using $I_1$ and $I_2$. Note that the models at each level are trained independently. At level 3, $I_3$ is an 8×8 image, simple enough to be modeled directly with a standard GANs $G_3$ & $D_3$.

Note that $G_k$ is a convnet which uses a coarse scale version of the image $l_k = u(I_{k+1})$ as an input, as well as noise vector $z_k$. $D_k$ takes as input $h_k$ or $\tilde{h}_k$, along with the low-pass image $l_k$ (which is explicitly added to $h_k$ or $\tilde{h}_k$ before the first convolution layer), and predicts if the image was real or generated. At the final scale of the pyramid, the low frequency residual is sufficiently small that it can be directly modeled with a standard GAN: $\tilde{h}_K = G_K(z_K)$ and $D_K$ only has $h_K$ or $\tilde{h}_K$ as input. The framework is illustrated in Fig. 2.

Breaking the generation into successive refinements is the key idea in this work. Note that we give up any "global" notion of fidelity; we never make any attempt to train a network to discriminate between the output of a cascade and a real image and instead focus on making each step plausible. Furthermore, the independent training of each pyramid level has the advantage that it is far more difficult for the model to memorize training examples – a hazard when high capacity deep networks are used.

As described, our model is trained in an unsupervised manner. However, we also explore variants that utilize class labels. This is done by add a 1-hot vector $c$, indicating class identity, as another conditioning variable for $G_k$ and $D_k$.

## 3 Model Architecture & Training

We apply our approach to three datasets: (i) **CIFAR10** [17] – 32×32 pixel color images of 10 different classes, 100k training samples with tight crops of objects; (ii) **STL10** [2] – 96×96 pixel color images of 10 different classes, 100k training samples (we use the unlabeled portion of data); and (iii) **LSUN** [32] – ∼10M images of 10 different natural scene types, downsampled to 64×64 pixels.

For each dataset, we explored a variety of architectures for $\{G_k, D_k\}$. Model selection was performed using a combination of visual inspection and a heuristic based on $\ell_2$ error in pixel space. The heuristic computes the error for a given validation image at level $k$ in the pyramid as $L_k(I_k) = \min_{\{z_j\}} ||G_k(z_j, u(I_{k+1})) - h_k||_2$ where $\{z_j\}$ is a large set of noise vectors, drawn from $p_{noise}(z)$. In other words, the heuristic is asking, *are any of the generated residual images close to the ground truth?* `Torch` training and evaluation code, along with model specification files can be found at `http://soumith.ch/eyescream/`. For all models, the noise vector $z_k$ is drawn from a uniform [-1,1] distribution.

### 3.1 CIFAR10 and STL10

**Initial scale:** This operates at $8 \times 8$ resolution, using densely connected nets for both $G_K$ & $D_K$ with 2 hidden layers and ReLU non-linearities. $D_K$ uses Dropout and has 600 units/layer vs 1200 for $G_K$. $z_K$ is a 100-d vector.

**Subsequent scales:** For CIFAR10, we boost the training set size by taking four $28 \times 28$ crops from the original images. Thus the two subsequent levels of the pyramid are $8 \rightarrow 14$ and $14 \rightarrow 28$. For STL, we have 4 levels going from $8 \rightarrow 16 \rightarrow 32 \rightarrow 64 \rightarrow 96$. For both datasets, $G_k$ & $D_k$ are convnets with 3 and 2 layers, respectively (see [5]). The noise input $z_k$ to $G_k$ is presented as a 4th "color plane" to low-pass $l_k$, hence its dimensionality varies with the pyramid level. For CIFAR10, we also explore a class conditional version of the model, where a vector $c$ encodes the label. This is integrated into $G_k$ & $D_k$ by passing it through a linear layer whose output is reshaped into a single plane feature map which is then concatenated with the 1st layer maps. The loss in Eqn. 2 is trained using SGD with an initial learning rate of 0.02, decreased by a factor of $(1 + 4 \times 10^{-4})$ at each epoch. Momentum starts at 0.5, increasing by 0.0008 at epoch up to a maximum of 0.8. Training time depends on the models size and pyramid level, with smaller models taking hours to train and larger models taking up to a day.

### 3.2 LSUN

The larger size of this dataset allows us to train a separate LAPGAN model for each of the scene classes. The four subsequent scales $4 \rightarrow 8 \rightarrow 16 \rightarrow 32 \rightarrow 64$ use a common architecture for $G_k$ & $D_k$ at each level. $G_k$ is a 5-layer convnet with $\{64, 368, 128, 224\}$ feature maps and a linear output layer. $7 \times 7$ filters, ReLUs, batch normalization [15] and Dropout are used at each hidden layer. $D_k$ has 3 hidden layers with $\{48, 448, 416\}$ maps plus a sigmoid output. See [5] for full details. Note that $G_k$ and $D_k$ are substantially larger than those used for CIFAR10 and STL, as afforded by the larger training set.

## 4 Experiments

We evaluate our approach using 3 different methods: (i) computation of log-likelihood on a held out image set; (ii) drawing sample images from the model and (iii) a human subject experiment that compares (a) our samples, (b) those of baseline methods and (c) real images.

### 4.1 Evaluation of Log-Likelihood

Like Goodfellow *et al.* [11], we are compelled to use a Gaussian Parzen window estimator to compute log-likelihood, since there no direct way of computing it using our model. Table 1 compares the log-likelihood on a validation set for our LAPGAN model and a standard GAN using 50k samples for each model (the Gaussian width $\sigma$ was also tuned on the validation set). Our approach shows a marginal gain over a GAN. However, we can improve the underlying estimation technique by leveraging the multi-scale structure of the LAPGAN model. This new approach computes a probability at each scale of the Laplacian pyramid and combines them to give an overall image probability (see Appendix A in supplementary material for details). Our multi-scale Parzen estimate, shown in Table 1, produces a big gain over the traditional estimator.

The shortcomings of both estimators are readily apparent when compared to a simple Gaussian, fit to the CIFAR-10 training set. Even with added noise, the resulting model can obtain a far higher log-likelihood than either the GAN or LAPGAN models, or other published models. More generally, log-likelihood is problematic as a performance measure due to its sensitivity to the exact representation used. Small variations in the scaling, noise and resolution of the image (much less changing from RGB to YUV, or more substantive changes in input representation) results in wildly different scores, making fair comparisons to other methods difficult.

| Model | CIFAR10 (@32×32) | STL10 (@32×32) |
|---|---|---|
| GAN [11] (Parzen window estimate) | -3617 ± 353 | -3661 ± 347 |
| LAPGAN (Parzen window estimate) | -3572 ± 345 | -3563 ± 311 |
| LAPGAN (multi-scale Parzen window estimate) | -1799 ± 826 | -2906 ± 728 |

Table 1: Log-likelihood estimates for a standard GAN and our proposed LAPGAN model on CIFAR10 and STL10 datasets. The mean and std. dev. are given in units of nats/image. Rows 1 and 2 use a Parzen-window approach at full-resolution, while row 3 uses our multi-scale Parzen-window estimator.

### 4.2 Model Samples

We show samples from models trained on CIFAR10, STL10 and LSUN datasets. Additional samples can be found in the supplementary material [5]. Fig. 3 shows samples from our models trained on CIFAR10. Samples from the class conditional LAPGAN are organized by class. Our reimplementation of the standard GAN model [11] produces slightly sharper images than those shown in the original paper. We attribute this improvement to the introduction of data augmentation. The LAPGAN samples improve upon the standard GAN samples. They appear more object-like and have more clearly defined edges. Conditioning on a class label improves the generations as evidenced by the clear object structure in the conditional LAPGAN samples. The quality of these samples compares favorably with those from the DRAW model of Gregor *et al.* [12] and also Sohl-Dickstein *et al.* [28]. The rightmost column of each image shows the nearest training example to the neighboring sample (in L2 pixel-space). This demonstrates that our model is not simply copying the input examples.

Fig. 4(a) shows samples from our LAPGAN model trained on STL10. Here, we lose clear object shape but the samples remain sharp. Fig. 4(b) shows the generation chain for random STL10 samples.

Fig. 5 shows samples from LAPGAN models trained on three LSUN categories (tower, bedroom, church front). To the best of our knowledge, no other generative model is been able to produce samples of this complexity. The substantial gain in quality over the CIFAR10 and STL10 samples is likely due to the much larger training LSUN training set which allows us to train bigger and deeper models. In supplemental material we show additional experiments probing the models, e.g. drawing multiple samples using the same fixed $4 \times 4$ image, which illustrates the variation captured by the LAPGAN models.

### 4.3 Human Evaluation of Samples

To obtain a quantitative measure of quality of our samples, we asked 15 volunteers to participate in an experiment to see if they could distinguish our samples from real images. The subjects were presented with the user interface shown in Fig. 6(right) and shown at random four different types of image: samples drawn from three different GAN models trained on CIFAR10 ((i) LAPGAN, (ii) class conditional LAPGAN and (iii) standard GAN [11]) and also real CIFAR10 images. After being presented with the image, the subject clicked the appropriate button to indicate if they believed the image was real or generated. Since accuracy is a function of viewing time, we also randomly pick the presentation time from one of 11 durations ranging from 50ms to 2000ms, after which a gray mask image is displayed. Before the experiment commenced, they were shown examples of real images from CIFAR10. After collecting ∼10k samples from the volunteers, we plot in Fig. 6 the fraction of images believed to be real for the four different data sources, as a function of presentation time. The curves show our models produce samples that are more realistic than those from standard GAN [11].

## 5 Discussion

By modifying the approach in [11] to better respect the structure of images, we have proposed a conceptually simple generative model that is able to produce high-quality sample images that are qualitatively better than other deep generative modeling approaches. While they exhibit reasonable diversity, we cannot be sure that they cover the full data distribution. Hence our models could potentially be assigning low probability to parts of the manifold on natural images. Quantifying this is difficult, but could potentially be done via another human subject experiment. A key point in our work is giving up any "global" notion of fidelity, and instead breaking the generation into plausible successive refinements. We note that many other signal modalities have a multiscale structure that may benefit from a similar approach.

## Acknowledgements

We would like to thank the anonymous reviewers for their insightful and constructive comments. We also thank Andrew Tulloch, Wojciech Zaremba and the FAIR Infrastructure team for useful discussions and support. Emily Denton was supported by an NSERC Fellowship.

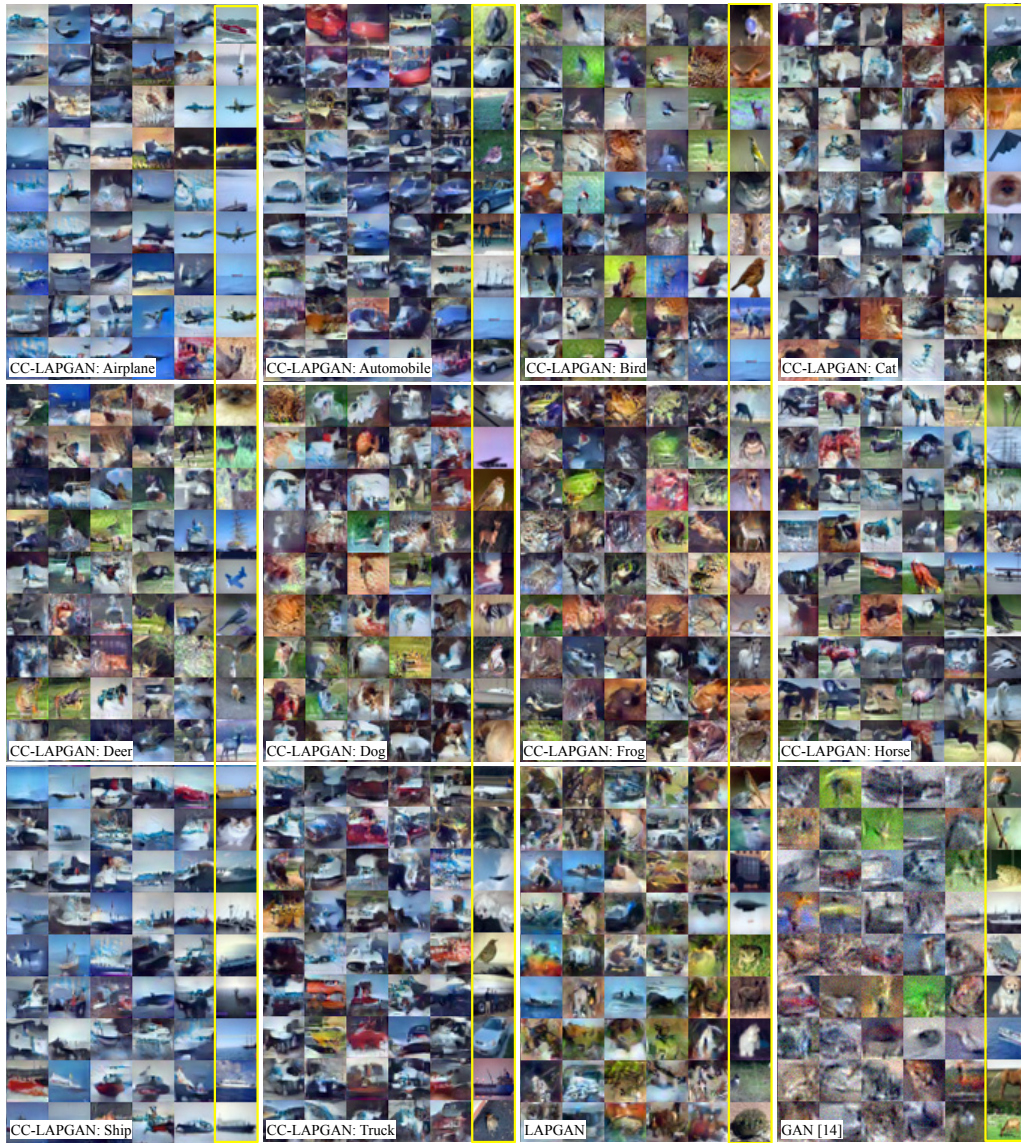

Figure 3: CIFAR10 samples: our class conditional CC-LAPGAN model, our LAPGAN model and the standard GAN model of Goodfellow [11]. The yellow column shows the training set nearest neighbors of the samples in the adjacent column.

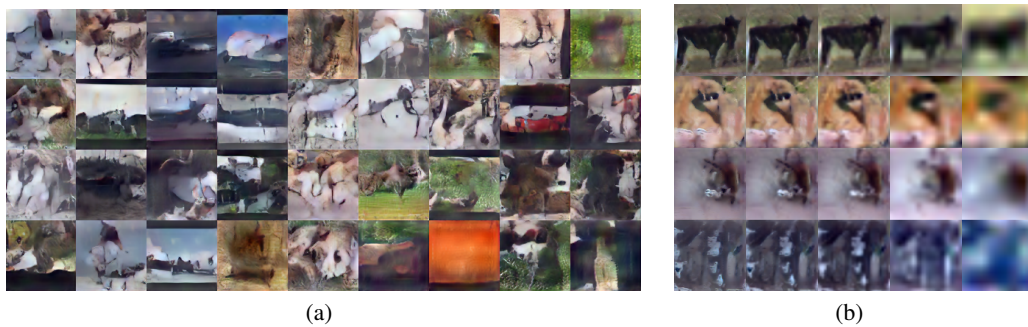

<table>
<tr><td>(a)</td><td>(b)</td></tr>
</table>

Figure 4: STL10 samples: **(a)** Random 96x96 samples from our LAPGAN model. **(b)** Coarse-to-fine generation chain.

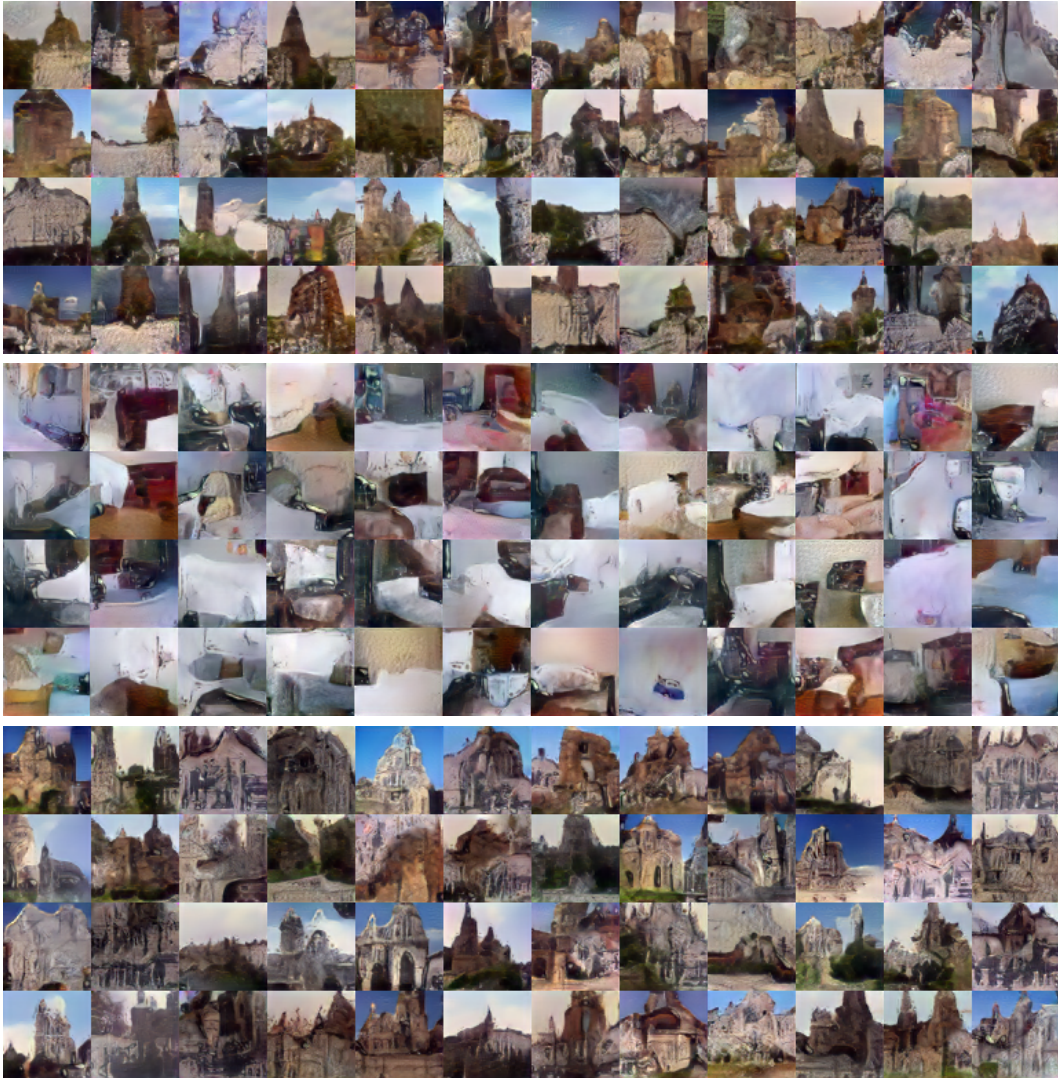

Figure 5: $64 \times 64$ samples from three different LSUN LAPGAN models (top: tower, middle: bedroom, bottom: church front)

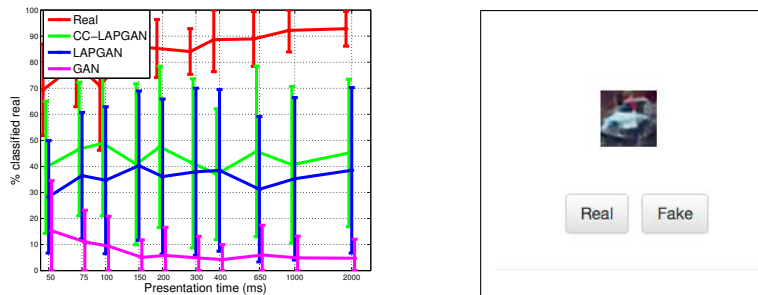

Figure 6: Left: Human evaluation of real CIFAR10 images (red) and samples from Goodfellow *et al.* [11] (magenta), our LAPGAN (blue) and a class conditional LAPGAN (green). The error bars show $\pm 1\sigma$ of the inter-subject variability. Around 40% of the samples generated by our class conditional LAPGAN model are realistic enough to fool a human into thinking they are real images. This compares with $\leq 10\%$ of images from the standard GAN model [11], but is still a lot lower than the $> 90\%$ rate for real images. Right: The user-interface presented to the subjects.

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
