[Supplementary Material · supp5.pdf]

# Deep Generative Image Models using a Laplacian Pyramid of Adversarial Networks
# Supplementary Material

**Emily Denton**[*]
Dept. of Computer Science
Courant Institute
New York University

**Soumith Chintala**[*]
Facebook AI Research
New York

**Arthur Szlam**

**Rob Fergus**

## Overview

Appendix A describes the multi-scale Parzen window estimator, used in Table 1 in the paper. Section 1 shows the architecture of models used for CIFAR10, STL10 and LSUN datasets. Section 2 shows additional samples and experiments on the LSUN dataset. Section 3 shows further samples from our CIFAR10 models.

## Appendix A

To describe the log-likelihood computation in our model, let us consider a two scale pyramid for the moment. Given a (vectorized) $j \times j$ image $I$, denote by $l = d(I)$ the coarsened image, and $h = I - u(d(I))$ to be the high pass. In this section, to simplify the computations, we use a slightly different $u$ operator than the one used to generate the images displayed in Figure 3 of the paper. Namely, here we take $d(I)$ to be the mean over each disjoint block of $2 \times 2$ pixels, and take $u$ to be the operator that removes the mean from each $2 \times 2$ block. Since $u$ has rank $3d^2/4$, in this section, we write $h$ in an orthonormal basis of the range of $u$, then the (linear) mapping from $I$ to $(l, h)$ is unitary. We now build a probability density $p$ on $\mathbb{R}^{d^2}$ by

$$p(I) = q_0(l, h)q_1(l) = q_0(d(I), h(I))q_1(d(I));$$

in a moment we will carefully define the functions $q_i$. For now, suppose that $q_i \geq 0$, $\int q_1(l) \, dl = 1$, and for each fixed $l$, $\int q_0(l, h) \, dh = 1$. Then we can check that $p$ has unit integral:

$$\int p \, dI = \int q_0(d(I), h(I))q_1(d(I)) dI = \int \int q_0(l, h)q_1(l) \, dl \, dh = 1.$$

Now we define the $q_i$ with Parzen window approximations to the densities of each of the scales. For $q_1$, we take a set of training samples $l_1, ...., l_{N_0}$, and construct the density function $q_1(l) \sim \sum_{i=1}^{N_1} e^{||l-l_i||^2/\sigma_1}$. We fix $l = d(I)$, and using this fixed $l$, we sample $N_0$ points $h_1, ..., h_{N_1}$ from the refinement model, and define

$$q_0(I) = q_0(l, h) \sim \sum_{i=1}^{N_0} e^{||h-h_i||^2/\sigma_0}.$$

Note that when defined this way, it is not obvious that $q_0$ is a measurable function, as the choice of $h_i$ by the upsampling model is different for every $l$ (and in fact depends on the random seed we used to sample). However, because the mapping from fixed "noise variable" and coarse image to refinement is the forward of a convolutional net, and so is continuous, if we use the same random seeds for each $I$, $q_1$ is measurable. For pyramids with more levels, we continue in the same way for each of the finer scales. Note we always use the true low pass at each scale, and measure the true high pass against the high pass samples generated from the model. Thus for a pyramid with $K$ levels, the final log likelihood will be: $\log(q_K(l_K)) + \sum_{k=0}^{K-1} \log(q_k(l_k, h_k))$.

---

[*]denotes equal contribution.

# 1 Model Architecture

Figure 1: Architecture of CIFAR10 and STL10 models.

Figure 2: Architecure of LSUN models.

# 2 LSUN

Fig. 3, Fig. 4 and Fig. 5 show samples drawn using the same $4{\times}4$ initial image (shown in leftmost column). Specifically, after generating from the 1st level GAN, the image is fixed and 8 different samples are then drawn, each using a different set of random noise vectors. These samples show that models produce plausible variations that cannot be the result of trivial copying of the training examples.

We can also condition the generation process on different coarse resolution images while keeping the noise vectors at each level fixed. Fig. 6(a), Fig. 6(b) and Fig. 6(c) show samples drawn from our tower, bedroom and church models respectively. The coarsest image (leftmost column) in the top and bottom rows of each figure were sampled from our 4x4 GAN. The intermediate coarse images were constructed by linearly interpolating between these two images. Each column shows a sample from a different level of the pyramid conditioned on the coarser image in the previous column. The same noise vectors were used for each row so that the only source of variation comes from the 4x4 images. An indication of overfitting would be the presence of sharp transitions in the generated images, despite the smoothly varying coarse input, as the model snaps between training examples. But this is not observed: the generations at each scale smoothly transition. Furthermore, each high resolution image looks like a plausible natural image, rather than a linear blend between two images. This indicates our model is moving along the manifold of natural images, rather than on a line between the start and end images.

# 3 CIFAR10

Fig. 7 shows additional samples drawn from our class conditional LAPGAN model, our LAPGAN model and the standard GAN model trained on CIFAR10. Fig. 8 shows nearest neighbors using L2 distance in pixel space of generated CIFAR10 samples samples. Fig. 9 shows nearest neighbors using L2 distance in feature space of a state-of-the-art convnet model[1], of generated CIFAR10 samples. These figure show that the model is not simply memorizing the training examples.

Figure 3: LSUN sample from class conditional LAPGAN model (tower) , seeded with generated $4 \times 4$ images (1st columns), with other columns showing different draws from the model.

Figure 4: LSUN sample from class conditional LAPGAN model (bedroom) , seeded with generated $4 \times 4$ images (1st columns), with other columns showing different draws from the model.

Figure 5: LSUN sample from class conditional LAPGAN model (church) , seeded with generated $4 \times 4$ images (1st columns), with other columns showing different draws from the model.

Figure 6: Effect of varying the coarsest input, with fixed noise at subsequent layers, on **(a)** tower model, **(b)** bedroom model and **(c)** church model.

Airplane

Cat

Frog

Automobile

Deer

Horse

Bird

Dog

Ship

Truck

Non-class conditional LAPGAN

GAN, Goodfellow et al. [14]

Figure 7: Samples drawn from our class conditional CIFAR-10 model.

Figure 8: Samples drawn from our class conditional CIFAR-10 model, with nearest neighbors in L2 pixel space shown in adjacent columns (orange).

Figure 9: Samples drawn from our class conditional CIFAR-10 model, with nearest neighbors in feature space shown in adjacent columns (orange).

## Footnotes

[1] Using this Network in Network model: https://gist.github.com/mavenlin/e56253735ef32c3c296d