[Reviews · NeurIPS 2015]

Submitted by Assigned_Reviewer_1

As in the summary -- the paper implements a sensible idea, it is clearly written, and the generated images are beautiful. I have a concern that the quantitative results are misleading and/or wrong.

I have a suspicion that the model may be performing similarly to a computer graphics engine ... where it generates very good naturalistic images, but where most real images would be assigned extremely low probability. This could make for a fine paper, but the results would need to be presented in a way that makes this clear.

Detailed comments as follow:

45 - 'indicating a better density model...' don't think this part follows

57-61 - Should note these aren't actually generative models as typically defined. Though p(visible|hidden) is straightforward, p(hidden) is complex and difficult to sample from.

72 - 'much more ambitious', and/or addresses a different goal

93 - 'their respective' -> 'both'

148 - prob. move footnote to main text, following a comma at the end of the sentence

186 - 'can directly' -> 'can be directly'

191-193 - Is this true? I would imagine fitting a model to each scale would just provide more parameters with which to overfit.

220 - I wonder if it will be difficult for these variables to represent something interpretable about the image, since there is only one noise variable per pixel, and the noise variables are interpreted convolutionally which will largely constrain them all to play a very similar role.

226 - On a validation set?

231 - Good idea + interesting to visualize variation captured.

239 - 'nature image' -> 'natural image'

Table 1: - Give units. Nats/image?

- If the units are nats/image, then I believe this performance would be state of the art if you compare against lossless image compression of 24-bit full size images. This would be impressive! But it also makes me extremely skeptical, because my understanding is that Parzen window techniques badly undershoot the true log likelihood, and 50,000 samples is a very small number of samples to fill the space of 32x32 images -- or even to fill a state space of dimensionality (32x32*3 - 16x16*3), after the conditioning. (e.g., assuming pixel values are integers, you would need order exp(1799 * (32*32*3 - 16*16*3) / (32*32*3)) samples to accurately capture the distribution using your Parzen window technique ... which is so much larger than 50000 that it leads to a floating point overflow when I try to compute it)

- Did you add uniform noise in [-0.5, 0.5] to each pixel value in your training and test sets? If not, it may be that scale 0 is learning to capture something about the integer quantization of pixel intensities. This is cheating, in that you can get infinite likelihood without learning anything about natural images by modeling the data as delta functions at integer pixel intensities. This could matter even for the multiscale Parzen approach, since it may be possible to exactly predict pixel values at scale 0 conditioned on scale 1.

- I'm especially suspicious that Parzen is being performed incorrectly, because you are also using non-comparable Parzen variants to compare GAN and LAPGAN. See Appendix A comment below.

- (A really neat followup project, if the numbers do turn out to be real, would be to use something similar to this method for image or video compression. I think the multi-scale structure would make for a straightforward encoding and automatically get you smooth performance degradation with limited bandwidth.)

- There are other models you could compare against here. [24] provides a bound on log likelihood for CIFAR-10. (and I believe also a deep learning workshop paper at this year's ICML -- will try to go back and look this up)

(Update: I realized from reviewer conversation that these log likelihoods are for 28x28 pixel images, not 32x32. That does explain the difficulty in direct comparison, and makes the log likelihood about 25% more believable. It would be good to note the image size in Table 1.)

Figure 5. - This is both interesting and difficult to know how to interpret. It seems as if image generation is quite stereotyped given initial 4x4 seeds. That would be consistent with the noise variables entering into later layers in a possibly difficult to utilize way (line 220). One interpretation might be that the top layer GAN creates most of the diversity in the generated samples, and that the conditional layers fill in consistent but inflexible details.

Section 5.3: - It's very difficult to know how subjects will choose to label something as "real" or "not real". For instance, if an image appears out of focus, or pixelated, or like an animal that you can't quite discern, do they still label it real? If you want to do a fair comparison against natural images, it would be much, much better to force subjects to make a choice between generated images and natural images. (this would be a "two alternative forced choice" experiment, and is extremely common)

- Who were the volunteers? What instructions did they receive? Did they know when they were looking at GAN vs. LAPGAN images, or what type of responses the experimenters hoped for in each session? Was the experiment single blind? Double blind? Did the subjects have previous experience with all three datasets? Strangers on mechanical turk, and a double blind experiment, would be better.

Some suggestions for two possible improved psychophysics experiments: 1) Do generated images look real: Show two images next to each other, and ask the user to select which one is 'real'. As they approach chance performance, this will be a strong indication that the images produced by the model appear realistic. 2) Are generated images diverse: Generating realistic images doesn't mean that you've captured the distribution over real images. For instance, maybe all the images you generate look very similar. One experiment that could significantly address this is to show real and generated images next to each other. Then show a third image, either real or generated, below, and ask which of the first two the third image is most similar to. The degree to which generated images are chosen to be similar to each other at above chance level is a measure of their diversity relative to natural images.

- If you are interested in the effect of exposure time, you should display a mask image (e.g. checkerboard or white noise) after the stimulus, rather than a gray screen. Otherwise your subject will have additional time to view the afterimage.

- You should provide a fixation target (e.g. a small cross) before each trial, and instruct the subject to fixate on it.

- This experiment seems obviously harmless, but you should be *very careful* about doing human experimentation without approval from an institutional review board. Running human experiments without review by an ethics board can get you in a lot of trouble, especially in a university setting. Many institutions have short online forms and a very easy expedited process for obviously harmless experiments like this one. Getting approval should be pretty lightweight, but is strongly recommended.

403 - 'previous scales' -> 'more than one previous scale' (assuming I've understood what's going on correctly)

Appendix A: This is a neat idea! Nice.

Parzen window estimates tend to be biased low because they require an exponential number of samples to fully explore a high dimensional space. Your method instead requires samples to explore a sequence of lower dimensional spaces. This will make it much easier for the samples to explore the state space, and can be expected to greatly increase the Parzen log likelihood, which is great.

You measure the GAN log likelihood using a Parzen estimate in the full space, and the LAPGAN log likelihood using your modified Parzen estimate. This isn't a fair comparison, since it is impossible to tell how much of the improvement is due to the improved Parzen estimation method, and how much is due to better performance of the model.

I strongly encourage you to include a third row in Table 1, where you give the LAPGAN log likelihood computed with the same Parzen estimator as the GAN row. Then it will be possible to tease apart model improvement vs. measurement technique improvement.

417 - notation, should probably make this q_0(h | l)

--

After reading the comments from Assigned_Reviewer_2, I share their concern that the number of latent variables is smaller than the number of free data dimensions in the conditional GAN layers (by a factor of 3 I believe). I would also like to request clarification on this.
Summary: This is a good idea, the samples are gorgeous, and the writing style was clear. I am pretty skeptical of the quantitative claims about performance.

The study appears to have used human subjects without approval by an ethics board. I don't know if NIPS has a formal policy on this, but for most journals this would preclude publication. The authors could almost certainly remedy this before the camera ready (e.g. by an expedited review at their institution followed by re-collection of data, or by removing that section of the paper).

Submitted by Assigned_Reviewer_2

Paper summary ============

This paper applies generative adversarial networks (GAN) to a Laplacian pyramid representation of natural images. The authors find that samples generated with the multiscale representation look more natural than samples generated by applying GANs directly to pixels.

Review summary =============

Using multiscale representations is a proven way to improve the performance of image models. I think that combining GANs with a Laplacian pyramid is a nice idea and the samples included in this paper suggest that this approach might also work well for GANs (but an alternative explanation is that the model overfits the data). Unfortunately, samples are an extremely unreliable test for the density estimation performance of a high-dimensional probabilistic model. Probabilistic models can have great samples and arbitrarily poor density estimation performance, or great density estimation performance while producing arbitrarily bad samples. Since the entire evaluation is based on samples (including the very limited Parzen window estimates), claims suggesting that the proposed approach represents "a significant advance" or a "better density model" are not supported by the experiments and should be removed*.

My biggest concern with this paper is that accepting it in the present form would further legitimize an evaluation of generative image models which is very easy to cheat. Other concerns relate to the meaningfulness and reproducibility of the comparison based on Parzen window estimates.

* I assume (as presumably would most people) that density estimation performance would be measured in terms of KL divergence. If the authors use a different definition of density estimation, then this definition needs to be given in clear terms in the paper.

Detailed review ============

It is trivial to construct a model which generates samples indistinguishable from real images by humans (e.g., simply store a number of training samples). To test for this kind of overfitting, the authors compare model samples to the closest examples in the training set (Figure 3). However, this test will only detect the simplest forms of overfitting, such as a true lookup table. This is because perceptually small changes can lead to big changes in Euclidean distance.

There are several reasons to believe that overfitting is still a real danger:

1) The goal of adversarial training is to generate samples which look like real samples to the discriminative network. Unlike for other well understood objectives, it is not clear how much a GAN is encouraged to have large entropy, which is important for generalization.

2) 100k images (CIFAR10, SL10) is still a very small amount of training data for density estimation in such high-dimensional spaces. What is the number of parameters used by each generative network?

3) Looking at Supplementary Figure 3, the variability of the samples created by conditioning on a low-resolution image is much smaller than what I would expect from real natural images. Some of the samples in main Figure 3 also look suspiciously similar (e.g. cars).

4) If I understood the description on page 5 correctly, the dimensionality of z_k is only the dimensionality of the (upsampled) low-pass image (e.g. 28 * 28 = 784), which is lower than what would be needed to model the RGB high-pass information with full support (e.g. 28 * 28 * 3 - 14 * 14 * 3 = 1,764). If this indeed the case, at least the KL divergence between the data and the model distribution is almost certainly infinite, since the model only captures a lower dimensional manifold (1,080 dimensions instead of 2,352 dimensions for CIFAR, if my calculations are correct).

If the model simply reproduced warped training images, that would explain all the results. Can you provide any evidence that the model is not simply doing that?

In Section 4.2 it sounds like the generative network is more complex than the discriminative network. How does changing the complexity of the discriminative network influence the results?

Where are the 100k training images for CIFAR10 coming from? If I remember correctly, CIFAR10 only contains 50k training images.

Although I appreciate the authors' efforts to quantify the model performance (Table 1), I doubt a Parzen window estimate will produce any meaningful results in such high dimensions (to their credit, the authors acknowledge the poor performance of Parzen window estimates). Please include more baselines (at least a Gaussian or ICA, which are very easy to train and evaluate) in the comparison. My hunch is that these would already produce much higher likelihoods.

I also see no reason not to compare to Sohl-Dickstein et al. (2015), who report log-likelihoods on CIFAR-10. Different preprocessing could be easily taken into account by adding a term for the Jacobian of the transformation, unless the preprocessing is non-invertible so that this is not possible. If it's due to the cropping to 28x28 pixels (the benefits of which are not entirely clear to me), the numbers should still be roughly comparable after normalizing by the pixel count. Either way, please explain the preprocessing in the paper. This is important for reproducibility and for enabling future comparisons.

It is furthermore not clear to me what we can learn from the quantitative comparison between GAN and LAPGAN, since the two models are evaluated with different approximations. Applying the Parzen window estimate to the two factors in $p(x) p(y | x)$ separately of course leads to higher likelihoods than applying it to the higher-dimensional $p(x, y)$ directly, as the authors will surely have observed by applying the Parzen window estimate to the LAPGAN samples in pixel space. But this says more about the inadequacy of the estimator than the performance of the model.

I realize that these models are difficult to evaluate (although obtaining a lower bound on the log-likelihood may be feasible, as shown by the work on auto-encoding variational Bayes). I also recognize that log-likelihood may not be the right criterion depending on the application. Previous work on intractable image models therefore demonstrated the models' usefulness in practice. E.g., by using them for denoising or pretraining neural networks. Since there does not seem to be an easy way to do inference in this model or even to evaluate an unnormalized likelihood, how is this model going to be used in practical applications?

Being able to produce nice looking samples might be of great use in image reconstruction applications. Unlike for density estimation, using human judgements for measuring image reconstruction performance would have been a welcome and indeed more meaningful alternative to the commonly used PSNR.

If the goal of the authors is indeed only to produce convincing looking images, the model should have been presented differently and compared to other image and texture synthesis algorithms (e.g., nonparametric sampling or Portilla & Simoncelli's algorithm). In that case it would have also been nice to include more quantitative analyses along the lines of texture similarity scores commonly used in texture synthesis. And how does LAPGAN relate and compare to 3D renderers? Note that there is currently also a lot of progress on performing inference in differentiable and probabilistic renderers (Kulkarni et al., 2015; Loper and Black, 2014).

Finally, what are the consequences of applying GAN to an overcomplete representation like the Laplacian pyramid (more dimensions than pixels)? Natural images should live on a lower dimensional manifold in this representation and degenerate distributions pose challenges for likelihood optimization. How do GANs solve this problem? Do the theoretical results of Goodfellow et al. (2014) carry over to this case?
Summary: I think that combining GANs with a Laplacian pyramid in general is a nice idea. However, I believe there are serious flaws in the presentation and the evaluation of the proposed model, so that I don't think the paper should be accepted in the present form.

Submitted by Assigned_Reviewer_3

The paper introduces an extension of generative adversarial networks (GAN) [10], specifically conditional GAN [9,16], for the purpose of learning a deep image model. The proposed LAPGAN model is based on a Laplacian pyramid decomposition of the image, where a separate (conditional) GAN model is trained at each level of the pyramid. Drawing a sample from the image model proceeds from coarse to fine, where the low-frequency components of the image are sampled first and higher-frequency parts of the image are sampled conditioned on the lower-frequency outputs. Compared to the baseline GAN model, the learned LAPGAN models are mainly evaluated based on their ability to produce realistic-looking samples, both visually and quantitatively via human evaluation.

# Positive It is a good and presumably novel idea to construct the image iteratively via successive models that capture the image in a coarse-to-fine approach. Furthermore, the proposed model seems like a substantial improvement over the baseline GAN model. The sampled images look quite realistic.

The paper is generally well-written and easy to follow. The related work is good and includes recent approaches. The experimental evaluation is quite comprehensive. I also appreciate the effort to quantitatively assess the improvement of the GAN model via human assessment.

# Negative I disagree with the sentence in l. 053-055: "For image processing tasks models based on marginal distributions of image gradients are popular [17, 21], but are only designed for image restoration rather than being true density models (so cannot sample an actual image)." One can actually sample images from these models [cf. Schmidt et al., "A Generative Perspective on MRFs in Low-Level Vision", CVPR 2010]. However, these sampled images can at best only reflect the marginal statistics that the model aims to capture. Otherwise, these samples look nothing like realistic images. Importantly, image models that only capture marginal image statistics can easily be used for images of arbitrary size and are often sufficient for applications like image restoration, where strong constraints are provided by a likelihood / data term. I would be interested to know whether the the (LAP)GAN model can actually be used for applications like image restoration? As far as I understand, this is not (easily) possible since the (LAP)GAN model is trained for images of fixed size. Is this correct? I am not disputing that generating realistic-looking images isn't interesting, but it might be a bit limited if that's the only application.

I would have liked a comparison to other (similar) image models, especially (class-)conditional GAN models [9,16]. The statement "The quality of these samples compares favorably with those from the DRAW model of Gregor et al. [11] and also Sohl-Dickstein et al. [24]." (l. 273-274) is unfortunately not backed up by any evidence.

(Update: After reading the other reviews, I am also skeptical of the quantitative evaluation of the proposed model. Please clarify.)

# Miscellaneous & typos - References missing for "Torch" and "Dropout", which are both mentioned first at the bottom of page 4. - The log-likelihood results in Table 1 are not discussed in the main text. - l. 112/113: "is constructed" -> "are constructed" - l. 050: "can in-painted" -> "can be in-painted" - l. 222: "who output" -> "whose output" - l. 403: "could remedied" -> "could be remedied"
Summary: The paper proposes a novel generative image model based on a pyramid of adversarial networks, which produces substantially more realistic looking samples than a

baseline model. It is unfortunately not clear if the proposed model has other applications besides generating

plausible images.

Author Feedback
Author rebuttal: We thank the reviewers for their constructive and lengthy reviews.

R2: "If the model simply reproduced warped training images, that would explain all the results. Can you provide any evidence that the model is not simply doing that?"

There is compelling evidence that this is not the case: (i) Fig. 5 shows compositional changes to the images (e.g. row 2, adding / deleting parts of towers) that cannot possibly be acheived via a warp; (ii) http://tinyurl.com/l2feats shows neighbours of our CIFAR10 samples computed in the feature space of a supervised convnet model with 13% error. The space is invariant to simple transformations and warps, yet the neighbors look dissimilar; (iii) similarly, http://tinyurl.com/nnlocal shows neighbors using a metric that permits *local* patch translations (Eqn. 3 in [A]). This allows highly complex transformations of the image, yet the neighbors are still far from our samples.

[A]: 80 million tiny images: a large dataset for non-parametric object and scene recognition, A. Torralba et al., TPAMI 2008.

R2: #1,#3 (large entropy / generalization of samples): This concern seems equally applicable to many existing generative models, e.g. samples from [11,24] also look rather similar to one another. Arguably, ours look more diverse than these, but since no objective measure exists, R2's concerns seem somewhat subjective.

R2: "an evaluation of generative image models which is very easy to cheat.": Lack of clear evaluation for generative models is widely acknowledged to be an issue in the field and this objection could be made equally of many recent papers. As explained above, it is clear that our model is not "cheating" by overfitting the training set. Despite its widespread use, log-likelihood (LLH) is a highly problematic measure (as discussed below), but we do use it to compare to other methods using the same Parzen window estimator, e.g. [10]. Additionally, our human subject experiment is an attempt to move beyond the subjective evaluation of samples.

R2: "Please include more baselines", compare to [24] (also R1) : The only way we can directly compare to e.g. [24] is using LLH (unnorm. bits/pixel -- ours: -20.5, [24]: -6.0, [10]: -23.0). However, LLH is problematic for degenerate distributions like images represented in pixel space. For example, a single Gaussian can get much higher LLH than on CIFAR than what we report or what is reported in [24] just because the covariance of the data is nearly degenerate. http://tinyurl.com/llhpcaf demonstrates this empirically: as suggested by R2, we fit a single Gaussian model fit to CIFAR training data with a varying number of PCA components, then evaluate the LLH of CIFAR test images under this model. The fig. shows the huge LLH values can be obtained, far higher than any published method. In light of this, it is hard to put too much credence on LLH as a performance metric.

However, we do attempt to compare with the approach in [10] because our model is so closely related, and our goal in including these LLH numbers is to show that our approach allows a refinement of the one in [10]. On the other hand, we do agree with the reviewers that it is not clear how much of the LLH gain over [10] is due to the estimation method as opposed to the model. We consider the ability to explore the product space, and thus generate many samples from the same low-pass to be the key advantage in our method; but will include a discussion of the distinction between gains from estimation vs. gains from modeling.

R2: "#params in gen. model": 8x8 model is fully connected and has ~5 million params. 8->14 and 14->28 are convnets with ~400k and ~800k params respectively.

R2 (&R1): #4: The extrinsic dim. of CIFAR10 images is 3072, the intrinsic dim. is far lower (a linear model with 784 DoF captures 99.3% of the energy). Thus our (non-linear) model is potentially able to capture virtually all the genuine image structure (as opposed to noise).

R2: "If I remember correctly, CIFAR10 only contains 50k training images...": Apologies, this is a typo (although flips and crops were used).

--

R1: "appears to have used human subjects without approval": The paper and experiments were performed at a company, where the labelers are employees whose contracts cover this kind of task. Also, the experiment passed internal legal review (which substitutes for a university IRB).
R1: Table 1 units: logprob/image.
R1: Sec 5.3: Thanks for the useful suggestions/questions -- we will clarify the text.

--

R3: "l.273-274 is unfortunately not backed up by any evidence": Please see Fig.3(d) in [24] and Fig.12 in [11]. Due to space limitations, they were not included in our paper.
R3: "comparison to... [9,16]": Neither of these papers address real image data.
R3: "Disagree with l.53-55": We will reword this. Our point is exactly what R3 says: "(Schmid et al.'s) samples look nothing like realistic images".